# Development of a Portable Dielectric Biosensor for Rapid Detection of Viscosity Variations and Its In Vitro Evaluations Using Saliva Samples of COPD Patients and Healthy Control

**DOI:** 10.3390/healthcare7010011

**Published:** 2019-01-16

**Authors:** Pouya Soltani Zarrin, Farabi Ibne Jamal, Niels Roeckendorf, Christian Wenger

**Affiliations:** 1IHP—Leibniz-Institut für innovative Mikroelektronik, Im Technologiepark 25, 15236 Frankfurt/Oder, Germany; 2farabi@gmail.com (F.I.J.); wenger@ihp-microelectronics.com (C.W.); 2Research Center Borstel—Leibniz Lung Center, 23845 Borstel, Germany; nroeckendorf@fz-borstel.de; 3Brandenburg Medical School, 16816 Neuruppin, Germany

**Keywords:** BiCMOS biosensors, COPD management and diagnosis, sputum–saliva characterization, dielectric measurements, radio frequency sensors

## Abstract

Chronic Obstructive Pulmonary Disease (COPD) is a life-threatening lung disease affecting millions of people worldwide. Although the majority of patients with objective COPD go undiagnosed until the late stages of their disease, recent studies suggest that the regular screening of sputum viscosity could provide important information on the disease detection. Since the viscosity of sputum is mainly defined by its mucin–protein and water contents, dielectric biosensors can be used for detection of viscosity variations by screening changes in sputum’s contents. Therefore, the objective of this work was to develop a portable dielectric biosensor for rapid detection of viscosity changes and to evaluate its clinical performance in characterizing viscosity differences of saliva samples collected from COPD patients and Healthy Control (HC). For this purpose, a portable dielectric biosensor, capable of providing real-time measurements, was developed. The sensor performance for dielectric characterization of mediums with high water content, such as saliva, was evaluated using isopropanol–water mixtures. Subsequently, saliva samples, collected from COPD patients and HC, were investigated for clinical assessments. The radio frequency biosensor provided high repeatability of 1.1% throughout experiments. High repeatability, ease of cleaning, low-cost, and portability of the biosensor made it a suitable technology for point-of-care applications.

## 1. Introduction

Chronic Obstructive Pulmonary Disease (COPD) is a life-threatening lung disease affecting millions of people worldwide [1]. It has been predicted that COPD will be the third leading cause of death in developed countries by 2030 [1]. The growth of COPD is significantly due to increase in tobacco use (including active smoking or secondhand smoke) and the air pollution [2]. Although many cases of COPD are considered to be treatable, early diagnosis is the key factor in their effective prevention and control [3]. COPD is an umbrella term describing chronic lung diseases that cause airflow constraints in lungs. The main symptoms of COPD are breathlessness, chronic cough, and abnormal sputum production [2]. Analyzing lung capacity of patients using a spirometer is the most common and rudimentary method for diagnosing COPD. However, sputum examination may be required for a more accurate diagnostics and identification of bacterial infections [4]. The sputum produced by lungs mainly consists of mucin, water, epithelial cells of the airway mucosa, and salts (in physiological concentrations). In addition, sputum could be contaminated with saliva during sampling. Since viscosity of sputum is mainly defined by its mucin (mucus glycoproteins) and water contents, regular monitoring of sputum viscosity could provide important information for early and fast detection of COPD [5,6]. However, it is not feasible to obtain sputum samples non-invasively in Point-of-Care (PoC) environments or from healthy controls on a daily basis. On the other hand, saliva is easily obtainable on a daily basis, in a non-invasive manner, with a better patient compliance. Therefore, in this work, we aimed to investigate whether viscosity of saliva samples is a proper biomarker for COPD, despite the fact that their viscosity differences are less distinct compared to sputum samples [6,7].

A broad range of technologies have been used for viscosity measurements of biofluids in medical applications [8]. Nonetheless, commercially available viscometers are mainly designed based on Micro Electro Mechanical Systems (MEMS). For example, Microfluidic Viscometer-Rheometer-on-Chip (m-VROC) is a MEMS-based pressure sensor array integrated into a microfluidic channel to measure the viscosity of Newtonian and non-Newtonian fluids. In this device, the test fluid is loaded in a micro-syringe and pumped inside the microfluidic. The pressure drop through the channel is measured by the sensor array and correlated to the fluid viscosity. The high accuracy and repeatability, small sample volume requirements, and the temperature control capability of the device make it an adequate technology for medical applications. However, its high cost, bulkiness, and cleaning complexity limit its use for PoC applications. Similarly, Microvisk (Microvisk Ltd., Oxfordshire, UK) is a commercially available viscometer for blood coagulation monitoring, functioning based on MEMS technology. In this device, piezoelectric elements pulsate steadily in the sample, characterizing its viscosity. The working principle of MEMS is based on the deflation of their cantilevers upon exposure to viscous fluid, which generates an electrical impulse. Many advantages of MEMS, such as high sensitivity and low cost, make them a well-established sensing technology for PoC [8,9,10]. However, their application for viscosity measurements is limited due to some drawbacks. The main limitation is the coupling between the fluid flow rate and its viscosity, impairing accuracy. In addition, the amount of drift, caused by resetting of sensor cantilevers back to their original position, is considerably high, which makes the calibration and accuracy of MEMS questionable in the long term. Moreover, a complicated cleaning process is required to remove stuck biological particles, such as protein molecules, from the sensor surface or capillary [10].

Apart from MEMS, other sensing technologies such as optical-sensors, pressure-sensors, and magnetic-sensors are moderately used for viscosity characterization of biofluids. However, their bulky size, operation–cleaning complexity, high cost, and lack of portability limit their application in real-world PoC devices [11,12,13,14,15,16]. In contrast, silicon-based technologies such as Complementary Metal Oxide Semiconductor (CMOS) have been widely used for the characterization of biological samples considering their high accuracy, miniaturization, robustness, and low cost [17]. These applications cover various dielectric spectroscopy methods for medical diagnosis and detection [18,19,20]. In addition, CMOS-based microorganism characterization using biochemical–biological markers is reported in the literature [21,22,23,24]. As an example, Stagni et al. developed a CMOS-based capacitive sensor for the detection of DNA molecules. In this work, the presence of DNA molecules are detected by capacitance variations of the sensor [23]. A similar detection principle was implemented to characterize various biological cell suspensions in organic fluids using dielectric biosensors [25]. It is shown that the dielectric characteristics of biological samples (blood, semen, and saliva) at microwave frequencies differ for patients with diseases compared to healthy subjects [20,26,27]. The relative permittivity (dielectric constant) of a medium is a dimensionless frequency-dependent complex number, the real ($\varepsilon_{r}^{\prime }$) and imaginary ($\varepsilon_{r}^{\prime \prime }$) parts of which represent a material’s energy absorption and energy loss, respectively, in an interaction with an electromagnetic field [26]. Therefore, the conductivity (σ), which accounts for the losses in a material, is definable using the following equation:(1)$$\sigma =\sigma_{ion}+f\varepsilon_{r}^{\prime \prime }\left(f\right)$$
where $\sigma_{ion}$ is the material’s ionic conduction and $\varepsilon_{r}^{\prime \prime }$ is the imaginary part of the permittivity at the functioning frequency of *f*.

Despite advancements in development of bio-viscometers, a reliable technology for screening the viscosity of sputum or saliva for rapid detection of COPD, at early stages, is still missing. Since viscosity variations of sputum, at different stages of COPD, are caused by changes in its water and mucin contents, dielectric sensors could potentially be used for screening viscosity changes for rapid monitoring of COPD. In our previous work, the concept of viscosity measurements using dielectric sensors was investigated [28,29,30]. Our previously developed BiCMOS dielectric sensor provided promising results for dielectric characterization of compounds such as ethanol, methanol, and isopropanol as well as mixtures (i.e., ethanol–methanol mixture). Additionally, dielectric characteristics of mixtures were correlated to their viscosity measures and the sensor’s capability for detecting viscosity changes was assessed [28,29,30]. However, the sensor integration into a portable device for PoC applications and its clinical performance evaluations were missing. Therefore, the objective of this work was to integrate our previously designed dielectric sensor into a handheld device for rapid characterization of saliva viscosity and to evaluate its in vitro performance for distinguishing viscosity differences of saliva samples of COPD patients and Healthy Control (HC). For this purpose, proper packaging of the system and development of a user-friendly interface were essential. In addition, a commercially available viscometer, with high accuracy and reliability, was required for the sensor performance evaluations during clinical trials. The following sections elaborate upon the technical design of the device and its clinical evaluations.

## 2. Sensor Design and Integration

As elaborately reported in our previous paper [28], the functioning principle of the sensor is based on the permittivity measurements of the Material-Under-Test (MUT) using capacitive elements (or more specifically four pairs of microstrip open-stubs), as shown in Figure 1a. These capacitive elements are coupled with inductors, defining the oscillation frequency of the LC resonant tank, Figure 1b. As a result, permittivity characteristics of the MUT alter the capacitance of the sensing elements, leading to a frequency shift in the resonator. As illustrated in Figure 1c, the frequency information of the oscillator is converted into DC signals, using a frequency discriminator, and extracted as the sensor output for the real part of the permittivity of the MUT. Moreover, the output power of the oscillator was detected using a power detector, providing information on the imaginary part of the permittivity of the MUT.

The sensor chip, with a 9.2 mm${}^{2}$ size and 80 mW power consumption, was fabricated using the 250 nm SiGe:C BiCMOS technology of IHP. The operation frequency of the sensor is in the range of 30 GHz, where a high signal-to-noise ratio is expected [19]. This is due to the fact that, based on the single Debye’s relaxation mechanism, the permittivity of water at 17 GHz is significantly high relative to other biological particles, making 10–30 GHz frequencies the most adequate range for dielectric spectroscopy applications [19]. In addition, the undesired parameter-dependent dispersion mechanism of biological cells, existing in low-frequency ranges, has negligible effects on the sensor measurements at its functioning frequency [19]. As shown in Figure 1d, DC readout, small size, and low power consumption of the sensor have made its fully integration into a handheld device possible.

Figure 2a illustrates the required parts for the full integration of the biosensor. The packaging of the device was fabricated out of a transparent resin (AR-M2) using a 3D printer (Keyence Agilista-3200W, Keyence Co., Osaka, Japan). The droplet reservoir, emplaced over the sensor area, was designed to access the MUT, while preventing sample spread over the Printed Circuit Board (PCB). Proper sealing of the reservoir was necessary for short-circuit prevention during handling conductive liquids, as shown in Figure 1a. Further details on sealing and packaging of the sensor are available in our previous work [28]. A 1.8” display (Raspberry PI, ST7735, SIMAC Electronics GmbH, Neukirchen-Vluyn, Germany) was used to provide measurement results to users, as shown in Figure 2b. In addition, an Arduino microcontroller (Mega 2560, SIMAC Electronics GmbH, Neukirchen-Vluyn, Germany) was used to supply DC power inputs, in order to acquire sensor outputs for post-processing, and to display the processed results on the interface display. Portability, ease of cleaning, low-cost, rapid detection possibility, and small sample requirements of the USB-powered device made it a suitable technology for home-care and PoC applications.

As shown in Figure 3a, a second version of the prototype, powered with four rechargeable batteries (1.2 V–1900 mAh, Fujitsu Ltd., Tokyo, Japan) and a simpler user interface (0.28” LED voltage panels, Seeed Technology Co., Shenzhen, China) was developed. Independency of this prototype from a USB power supply makes it a suitable technology for remote applications. The laptop-shape design of the packaging secures the sensor surface in remote–harsh environments, as shown in Figure 3b.

## 3. Experimental Setup

### 3.1. Mixture Detection

As reported in our previous paper, the sensor provided an accuracy of 4.17% for the dielectric characterization of low-conductive liquids such as ethanol [28]. In addition, it was able to accurately distinguish ethanol–methanol mixtures with various methanol concentrations. However, considering water as the major component of saliva [31], it was critical to evaluate the sensor performance in characterizing conductive mixtures prior to its clinical assessment. Water is a highly conductive lossy material that impairs the dielectric sensor’s function.

For this purpose, mixtures of isopropanol–water (deionized water), ranging from 0–100% water (in volumes), were prepared and used for the sensor performance assessments. The effective permittivity of these mixtures at different steps were calculated using mixture theories [32]. The effective permittivity ($\varepsilon_{r,eff}$) of a binary isotropic mixture, with mixing ratios of 1 −*q* and *q* for its host (the dominant content, $\varepsilon_{r,h}$) and guest ($\varepsilon_{r,g}$) constituents, is calculable using the following equation:(2)$$\varepsilon_{r,eff}=\frac{\varepsilon_{r,g}\left(2q+1\right)+2\varepsilon_{r,h}\left(1-q\right)}{\left(2+q\right)+\frac{\varepsilon_{r,g}}{\varepsilon_{r,h}}\left(1-q\right)}$$

Debye-based relaxation equations, for modeling the frequency dispersive behavior of materials, were required to calculate the complex permittivity of water and isopropanol at 30 ${}^{\circ }$C and 27 GHz for the operating temperature and frequency of the sensor, respectively [33,34,35]. The double-Debye relaxation equation, with high accuracy for modeling materials with two dielectric relaxations, was used for the complex permittivity ($\varepsilon^{*}=\varepsilon_{r}^{\prime }-j\varepsilon_{r}^{\prime \prime }$) calculations, Equation (Equation 3).
(3)$$\varepsilon^{*}=\varepsilon_{\infty }+\frac{\varepsilon_{s}-\varepsilon_{h}}{1+jf/f_{r1}}+\frac{\varepsilon_{h}-\varepsilon_{\infty }}{1+jf/f_{r2}}$$

In this equation, $f_{r1}$ and $f_{r2}$ are the relaxation frequencies of the material’s two dielectric relaxations. The high-frequency permittivity limit of the material is $\varepsilon_{\infty }$; however, $\varepsilon_{s}$ and $\varepsilon_{h}$ are representing the static permittivity and the notional high-frequency permittivity limit of the lower frequency relaxation, respectively. The calculated theoretical values for the real ($\varepsilon_{r}^{\prime }$) and imaginary ($\varepsilon_{r}^{\prime \prime }$) parts of the complex permittivity of these mixtures were recorded and compared to the biosensor results, as presented in Figure 4 and Figure 5.

Figure 4 shows the sensor output for the real ($\varepsilon_{r}^{\prime }$) part of the effective permittivities of mixtures and their theoretical values. It is shown that, by adding water into mixtures, both the output voltage of the sensor and the dielectric constant values of mixtures increase. This is along with the functioning principle of the sensor, since increasing the water content of mixtures (higher dielectric constant) triggers the input capacitance of the sensor, leading to a higher output voltage. However, after the 50–50% step, water becomes the dominant content of the mixture, stopping the sensor from functioning adequately.

The sensor outcome for the imaginary ($\varepsilon_{r}^{\prime \prime }$) part of the effective permittivities of mixtures and their theoretical values are illustrated in Figure 5. Increase in the water content (higher conductivity) of mixtures leads to a greater energy leakage and a decline in the output power of the oscillator. Similar to acquired results for $\varepsilon_{r}^{\prime }$, the sensor stops functioning properly after the water dominance in the mixture. This is due to the fact that, after the 50–50% step, the energy leakage in the system is so high (so lossy) that the sensor cannot track the input changes accordingly and, consequently, the sensor output remains constant (Figure 5).

### 3.2. Clinical Evaluations

An m-VROC viscometer (m-VROC, RheoSense Inc., San Ramon, CA, USA), capable of viscosity characterization of non-Newtonian biofluids, was used for the sensor performance evaluations. Two groups of saliva samples, collected from HC and COPD patients (five samples for each group), were de-frozen and prepared for measurements, as illustrated in Figure 6a. It should be noted that the sampling of the saliva samples was approved by the local ethics committee of the University of Luebeck (approval no. 16-167) and a written informed consent was obtained from all patients. A commercialized centrifuge (Eppendorf centrifuge 5415R, Eppendorf Inc., Hamburg, Germany) was used to separate and remove particulate pallets from saliva samples, as shown in Figure 6b. The process was conducted at 4 ${}^{\circ }$C and 4000 RPM for 5 min. After the centrifugation process, the viscosity and complex permittivity of samples were simultaneously measured using m-VROC and the dielectric sensor, respectively. The experiments were conducted in triplicate following cleaning (using Aquet) and primary steps required for m-VROC measurements. The average values of three measurements were calculated and reported for every sample, as shown in Figure 7. During the last two sets of experiments, the micro-capillary of m-VROC was blocked by particulate matters existing in saliva (such as protein molecules). As a consequence, m-VROC viscosity results for these measurements were inaccurate and, therefore, were excluded from calculations and the report, as shown in Figure 7. The inaccuracy and unreliability of these measurements were evident considering their low r-square values, provided for their slope-fit by m-VROC, as well as their irrelevancy with respect to the expected viscosity range for saliva samples (1.5 mPa·s) [7].

Figure 8 and Figure 9 show the biosensor results for the complex permittivity measurements of saliva samples collected from COPD patients and HC. The real part of the permittivity indicates the energy absorption capability of these two groups of samples; however, the imaginary part of the permittivity represents the energy loss differences in them. Moreover, by taking Equation (Equation 1) into account, the imaginary part of the permittivity (energy loss) could indicate conductivity characteristics of samples.

## 4. Results and Discussion

Table 1 presents the results of the complex permittivity and viscosity measurements acquired by the dielectric biosensor and m-VROC for COPD patients and HC. The relative standard deviation method was implemented on the acquired results to determine the repeatability characteristics of sensors. Repeatability values of 1.3% and 1.1% were calculated for m-VROC and the biosensor, respectively.

Viscosity measurements by m-VROC provided important information on the physical properties of saliva samples of COPD patients and HC, beneficial for disease diagnosis. As shown in Figure 7, the average viscosity of saliva samples is approximately 0.15 mPa·s greater for COPD patients compared to HC. However, considering this extremely small difference, it is crucial to take into account the effects of measurement conditions on the results. For example, temperature fluctuations can significantly affect the sensor performance. Therefore, temperature compensation methods are necessary to improve the sensor accuracy for real-world applications. Furthermore, the sample fraction from which the saliva was collected (after the centrifugation process) determines its density level. As a result, saliva samples from different fractions might show slightly different viscosity characteristics.

On the other hand, the developed biosensor provided valuable information on the electrical properties of saliva samples of COPD and HC groups by distinguishing between their complex permittivity values, as illustrated in Figure 8 and Figure 9. As shown in Figure 8, the average sensor output for the real part of the permittivity were shown to be 0.65 (V) and 0.58 (V) for COPD and HC, respectively. Therefore, COPD samples provided better dielectric characteristics (energy absorption capability) compared to HC. However, considering the biosensor results for the permittivity characterization of isopropanol–water mixture (Figure 4), the measured values for the real part of the permittivity of COPD samples are out of the expected measurement range. This issue could be caused due to the presence of highly dielectric materials such as salt or other biological particles in COPD samples. Therefore, further investigations from a biological point of view are required to identify the main cause behind this phenomenon, which is out of the scope of this paper.

As illustrated in Figure 9, the average sensor output for the imaginary part of the permittivity of COPD and HC were shown to be 0.071 (V) and 0.064 (V), respectively. Due to a greater energy leakage (energy loss) in HC samples, the imaginary part of the permittivity of these samples is lower than COPD. As discussed in Figure 5, lower values for the imaginary part of the permittivity are representing higher conductivity features. Therefore, saliva samples of HC presented better conductivity characteristics compared to COPD, which could be due to the presence of a greater water content in HC samples. Since the greater water content of saliva represents its lower viscosity level, sensor results suggest higher viscosity values for COPD samples relative to HC. This conclusion goes along with the viscosity results acquired from m-VROC.

Although no direct theoretical correlation between the viscosity (mechanical properties) and the dielectric constant (electrical properties) of a medium exists, m-VROC and the biosensor were able to identically track the variations of these two features in saliva samples of COPD and HC. In other words, the dielectric biosensor was able to detect viscosity differences between saliva samples of COPD and HC by identifying their complex permittivity differences. However, considering the functioning principle of a dielectric sensor, providing absolute viscosity values of these samples was not possible.

Considering the high water content of saliva and the notably small viscosity differences of COPD and HC samples, detection of dielectric constant variations in these samples requires an extremely accurate and sensitive sensing technology. Therefore, modifications are required to improve the accuracy and sensitivity of the dielectric sensor for characterization of these kinds of samples. Since the developed biosensor is able to characterize dielectric properties of a MUT from millimeter distances, locating the sensor a few millimeter distanced from the droplet reservoir could potentially address the issues related to the extensive energy loss caused by highly conductive liquids such as water. However, the trade-off between the sensor resolution and its distance from the MUT need to be taken into account. In addition, using bottom fractions of the centrifuged saliva (the high-density region with less water) for measurements could possibly improve the sensor performance.

It is noteworthy that the goal of this study was to show the feasibility of the concept in a preclinical setting. However, further investigations, with a large number of saliva samples (preferably 50 or more), are required in the future to show the precise and accurate trend in viscosity variations for COPD and HC groups. Upon the unavailability of clinical samples in such a large number, artificially simulated saliva samples could be used for these investigations. However, it should be taken into account that the spiking volume of the external substances, added for increasing the viscosity of specific samples, should not exceed 5% of the artificial saliva. In addition, it needs to be mentioned that the variability of saliva as a sample matrix, due to various factors such as a patient’s diet, could potentially affect viscosity properties of samples and, consequently, the accuracy of clinical assessments. Therefore, a standard operating procedure (SOP) was implemented in our sampling method to reduce potential differences in our sample pool and in between samples. However, viscosity differences of saliva samples cannot be the one and only parameter for determining the absolute state of the disease, but a broader panel of COPD biomarkers, such as cytokine levels and pathogen load of samples, is required for an accurate diagnosis. As a result, the low reliability level of saliva, considering its variability, is among the main caveats of this assay procedure for clinical assessments. On the other hand, main advantages of using saliva as a sample matrix are its availability in PoC environments and the possibility of obtaining samples non-invasively on a daily basis. Future enhancements required for the efficiency of the testing procedure include using artificially simulated saliva and modifying the cleaning–calibration processes of the system for conducting large number of experiments in a shorter period of time.

A major advantage of the developed biosensor was its capability for static characterization of non-Newtonian liquids with no fluid flow requirements. This feature of the device made its cleaning process relatively faster and simpler compared to microfluidic-based systems such as m-VROC, in which removing protein molecules stuck inside its capillary is very challenging and time consuming. Moreover, the cost-effectiveness of the system, due to its simplified design and packaging, as well as its notably short response time are among its main advantages compared to other emerging technologies. Considering these advantages, the biosensor could potentially be used in the future for other bioanalytical applications and biomarkers detection such as micro-organisms characterization, medical spectroscopy, glucose concentration monitoring, cerebra spinal fluid characterization, DNA detection, and sperms identification for semen analysis.

As previously mentioned, ambient conditions such as temperature can notably affect the samples viscosity characteristics and, consequently, the biosensor outcome. However, this issue could be problematic from a medical–diagnostic point of view. To address this issue, acquired results from the biosensor need to be sent to a web-enabled (cloud-connected) smart device, such as a smartphone or a tablet, for further comparison with available patient records. Development of diagnostic algorithms based on different ambient conditions is essential for this purpose. Training neural networks for optimizing these diagnostic algorithms using available medical records will pave the way towards an artificial intelligence-based diagnostic method. Nevertheless, this complex method requires further investigations and is considered as future work.

## 5. Conclusions and Future Work

In this work, a dielectric biosensor was developed for the rapid detection of viscosity variations of biofluids. The sensor was fully integrated into a handheld device and provided rapid and real-time measurements for users. Low-cost, simplified packaging, ease of cleaning, portability, and rapid detection capability of the biosensor made it a suitable technology for PoC applications.

The sensor efficacy for the dielectric characterization of water-based mediums was assessed using isopropanol–water mixtures. The sensor showed promising results for complex permittivity characterizations of mixtures up to the point that water dominates the solutions.

The in vitro performance of the biosensor was evaluated using a commercialized m-VROC viscometer. Viscosity and complex permittivity of saliva samples, collected from COPD patients and HC, were characterized and the capability of the biosensor to detect their viscosity differences was evaluated. Throughout these experiments, COPD samples provided higher viscosity compared to HC. Similarly, greater values for the real part of the complex permittivity of COPD samples, representing their better dielectric features, were recorded. In contrast, the imaginary part of the permittivity of HC, which indicates conductivity characteristics, were shown to be greater compared to the COPD group. Further investigations are required to clarify the biological causes on the obtained results.

The biosensor showed promising results for the detection of viscosity differences of saliva samples of COPD and HC through detecting their permittivity variations. In addition, the capability of the biosensor to distinguish between the electrical properties of COPD and HC saliva samples, opened up a new window into different methodologies for early and rapid detection of COPD. High repeatability of 1.1% was reported for the biosensor, making it a reliable technology for medical diagnostics. Nevertheless, future modifications are required to compensate temperature effects on measurements and to make the biosensor more compatible to mediums with a high water content. In addition, considering the recent trend in medical technologies towards the internet of things for smart healthcare systems [36,37], the next generation of the biosensor will be interfaced with a smartphone to enable physicians with a continuous monitoring of the disease progress. Furthermore, by sending the collected data to the cloud, it will be possible to implement machine learning methods for precision-diagnostic purposes. The above-mentioned applications of the biosensor for further medical diagnostics will be investigated in the future.

## Figures and Tables

**Figure 1 healthcare-07-00011-f001:**
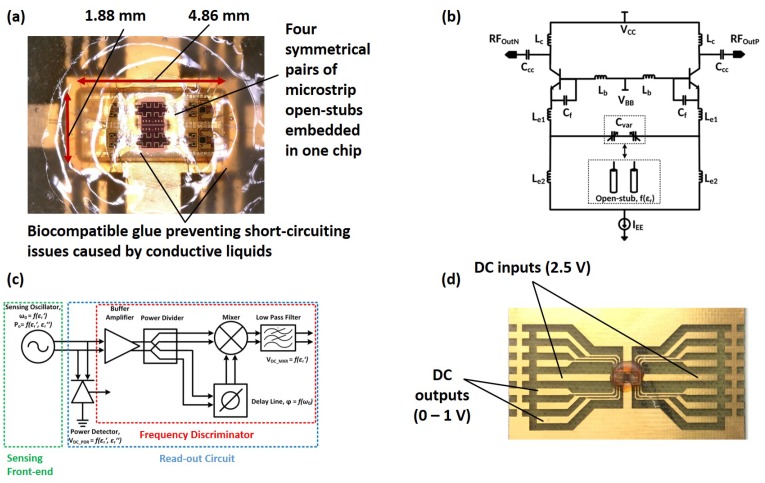
(**a**) capacitive elements, four pairs of microstrip open-stubs, used for permittivity characterization of the MUT; (**b**) sensor oscillator circuit, illustrating inductors coupled with the capacitive elements; (**c**) DC read-out schematic, elaborating the frequency discriminator and the power detector; and (**d**) DC inputs and outputs for the sensor chip embedded on a PCB.

**Figure 2 healthcare-07-00011-f002:**
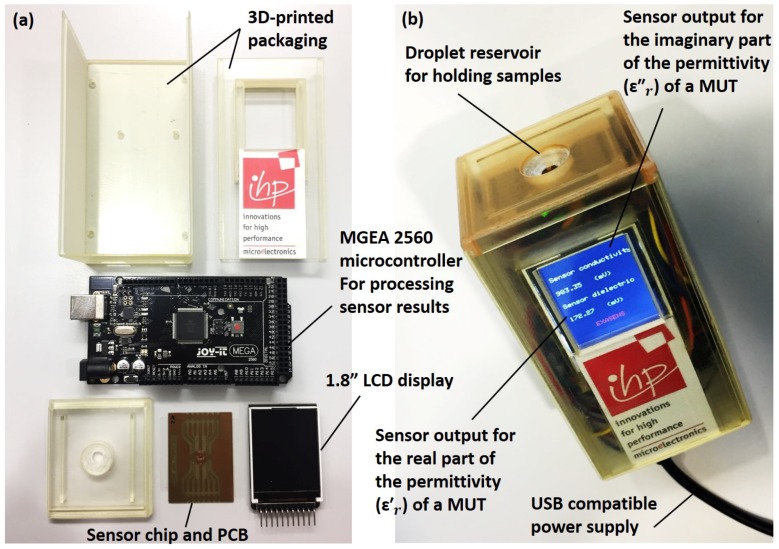
(**a**) various parts required for the fully integration of the biosensor into a handheld device including a microcontroller, an LCD display, and a 3D-printed packaging; (**b**) assembled device working with a USB power supply.

**Figure 3 healthcare-07-00011-f003:**
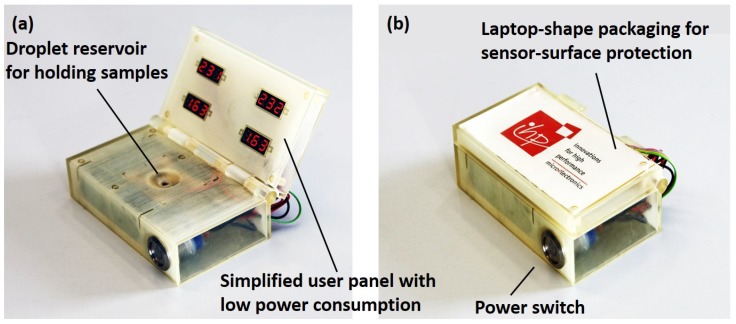
(**a**) battery-powered version of the biosensor suitable for harsh and remote environments with a limited access to a USB power supply; (**b**) laptop-shaped foldable packaging suitable for sensor-surface protection.

**Figure 4 healthcare-07-00011-f004:**
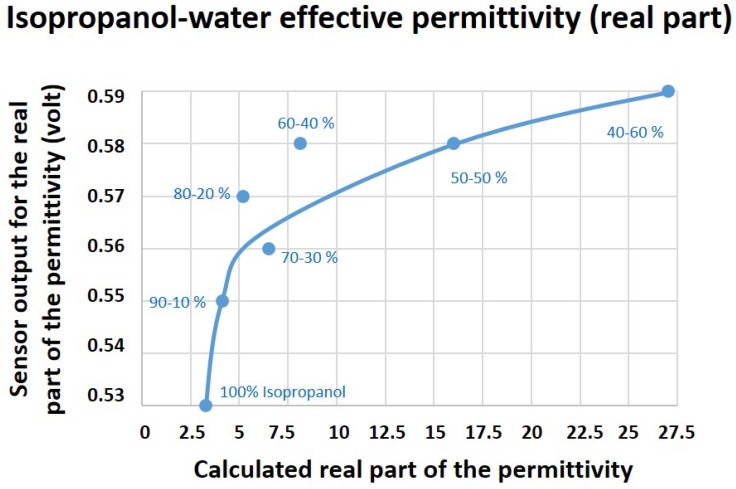
Sensor results for the real ($\varepsilon_{r}^{\prime }$) part of the effective permittivity of isopropanol–water mixtures compared to its theoretical values calculated using mixture theories.

**Figure 5 healthcare-07-00011-f005:**
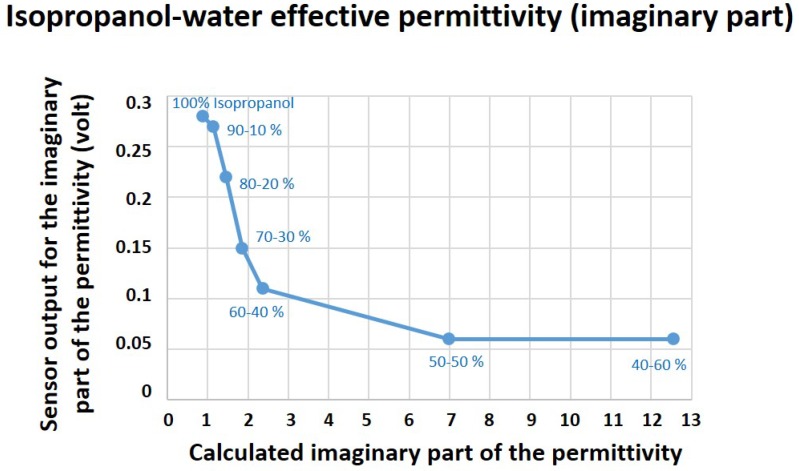
Sensor results for the imaginary ($\varepsilon_{r}^{\prime \prime }$) part of the effective permittivity of isopropanol–water mixtures compared to its theoretical values calculated using mixture theories.

**Figure 6 healthcare-07-00011-f006:**
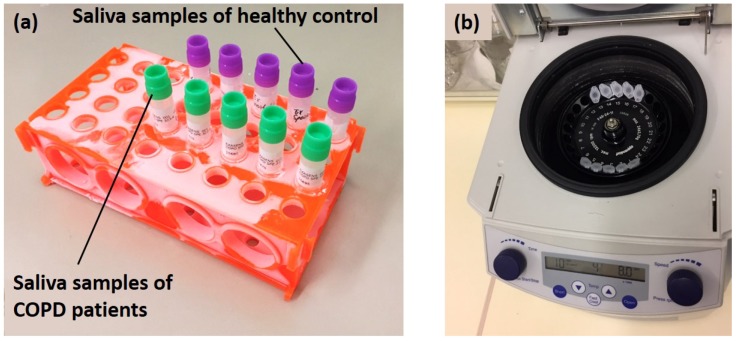
(**a**) two groups of saliva samples collected from HC and COPD patients (five samples for each group); (**b**) centrifuging saliva samples for particulate pallets separation and removing.

**Figure 7 healthcare-07-00011-f007:**
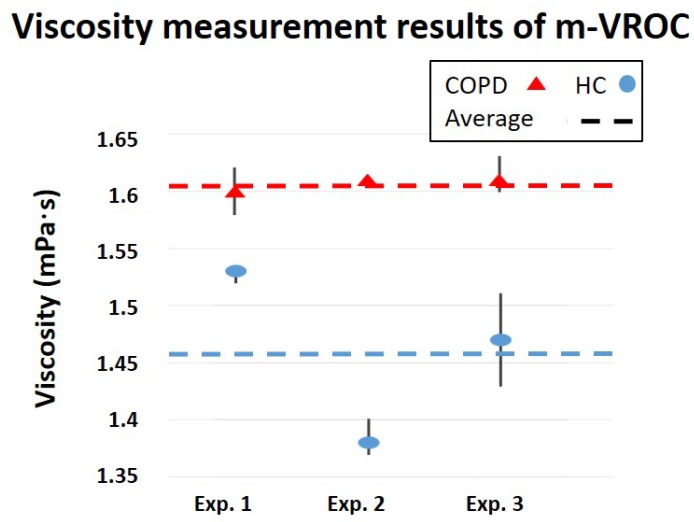
Results of m-VROC viscosity measurements for saliva samples of COPD patients and HC.

**Figure 8 healthcare-07-00011-f008:**
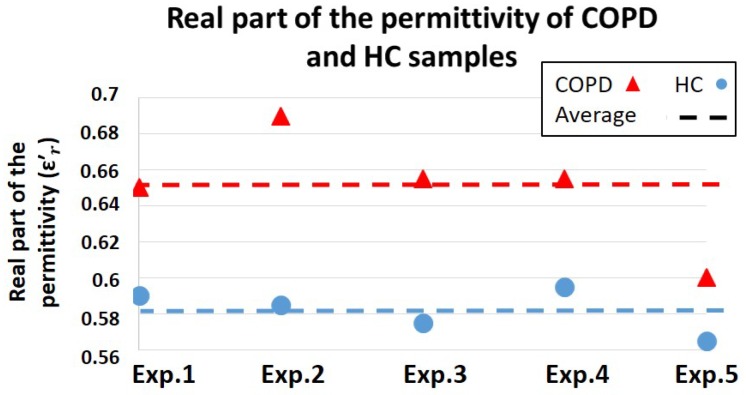
Results of the biosensor measurements for the real part of the permittivity indicating the energy absorption capability of saliva samples of COPD and HC.

**Figure 9 healthcare-07-00011-f009:**
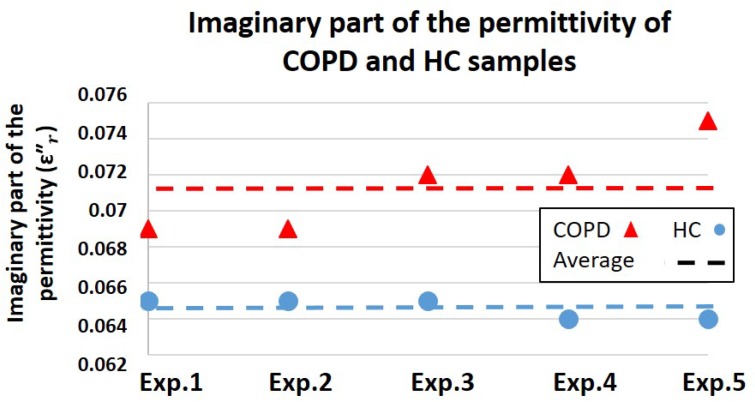
Results of the biosensor measurements for the imaginary part of the permittivity represents the energy loss differences in COPD samples compared to HC.

**Table 1 healthcare-07-00011-t001:** Results of the complex permittivity and viscosity measurements for COPD and HC patients.

Samples	m-VROC (mPa·s)	Biosensor $\varepsilon_{r}^{\prime }$ (Volt)	Biosensor $\varepsilon_{r}^{\prime \prime }$ (Volt)
COPD	1.61	0.65	0.071
HC	1.46	0.58	0.064

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
