# Peer review of "Development of a Portable Dielectric Biosensor for Rapid Detection of Viscosity Variations and Its In Vitro Evaluations Using Saliva Samples of COPD Patients and Healthy Control"

_healthcare, 2019, doi:10.3390/healthcare7010011_

Round 1

Reviewer 1 Report

The authors have reported an interesting dielectric biosensor for the detection of viscosity variations, which has been demonstrated for the analysis of COPD patient's saliva samples.

Although the concept is good and the results correlates with the predicate technique for visualizing the trend in viscosity variations, there are some minor concerns that need to be addressed.

The clinical evaluation of COPD samples should have been done with a large number of samples more than 50 to show that the biosensor is showing precise and accurate trend in viscosity variations.

If the clinical samples are unavailable, the authors can also use simulated artificial saliva samples and change its viscosity by adding external substances taking into account that the spiking volume should not exceed 5% of artificial saliva.

The viscosity of saliva in healthy controls could be changed by many factors including diet. Moreover, saliva is one of the most variable clinical sample matrix. Therefore, the authors should comment on the pros and cons of using saliva only for COPD sample analysis.

Could the developed sensor be useful for other bioanalytical applications and other sample matrices? The authors should report all possible future applications of their biosensor in a section before conclusions.

The cost-effectiveness and response time should be compared w.r.t. to established and emerging central lab and POCT technologies so that the readers are clear. The various technologies should be compared in terms of their characteristics in a table.

The effect of ambient conditions on the sensor response should also be analyzed.

Perhaps, if the biosensor could be interfaced with smart devices such as smartphone, this could lead to next-generation biosensor. The authors should comment on this briefly and should add some relevant refs. here such as Trends Biotechnol, 33(11), 692-705, 2015.

Author Response

The authors would like to thank the Associate Editor and the reviewers for their constructive comments and suggestions concerning the above-mentioned paper. The paper has been revised to address their comments and suggestions. 

Reviewer 2 Report

The work by Zarrin et al., describes a novel dielectric biosensor for detection of COPD by assessing the viscosity variations in sputum samples. The study is well designed and presented. The study has potential implication in COPD point-of-care diagnosis. Only minor correction to be made is to discuss the caveats of this assay procedure and future enhancements to make the test more efficient. This part jeds to be included in the discussion part. 

Author Response

(The authors gave the same response as above.)
